# Detection of Antibiotic Resistance, Virulence Gene, and Drug Resistance Gene of *Staphylococcus aureus* Isolates from Bovine Mastitis

Zhe Zhang,[a] Yun Chen,[a] Xinpu Li,[a] Xurong Wang,[a] (ID) Hongsheng Li[a]

[a]Lanzhou Institute of Husbandry and Pharmaceutical Sciences, Chinese Academy of Agricultural Sciences/KeyLab of Veterinary Pharmaceutical Development, Ministry of Agriculture and Rural Affairs/Engineering and Technology Research Center of Traditional Chinese Veterinary Medicine, Lanzhou, People's Republic of China

Zhe Zhang and Yun Chen contributed equally to this work. Author order was determined on the basis of seniority.

**ABSTRACT**   Antimicrobial therapy plays an important role in mastitis control caused by *Staphylococcus aureus* but has become less effective due to widespread drug resistance. The purpose of this study was to detect antibiotic resistance, drug resistance gene, and virulence gene of *S. aureus* strains. In this study, 2,962 milk samples were collected from 43 dairy farms located in 16 provinces of China and cultured for isolation of *S. aureus*. Antibiotic resistance, capsular polysaccharide, spa typing, virulence genes, and drug resistance genes of the strains were analyzed. Of 2,962 samples, 298 strains were isolated and identified as *S. aureus*. The strains exhibited high percentages of resistance to penicillin G (91.95%). Moreover, all strains showed resistance to more than one antimicrobial agent but were sensitive to nitrofurantoin and sulfamethoxazole/trimethoprim. The results indicate that type 8 was the dominant capsular polysaccharide serotype and t459 was the dominant spa type. The most prevalent virulence gene was *clfA* (98%). The resistance genes of several antibiotics were detected, among which the *blaZ* gene (92.95%) was the highest. In conclusion, we present the antimicrobial resistance and virulence genes of *S. aureus* in this study which are of importance for mastitis control.

**IMPORTANCE** Bovine mastitis is a serious disease associated with both high incidence and economic loss, posing a major challenge to the dairy industry worldwide. *Staphylococcus aureus* is one of the most common pathogens to cause bovine mastitis, and antimicrobial therapy plays an important role in mastitis control caused by *S. aureus* but has become less effective due to widespread drug resistance. The purpose of this study was to detect antibiotic resistance, drug resistance gene, and virulence gene of *S. aureus* strains, which would be helpful to mastitis control.

**KEYWORDS** *Staphylococcus aureus*, antibiotic resistance, virulence gene, drug resistance gene

**B**ovine mastitis is a serious disease associated with both high incidence and economic loss, posing a major challenge to the dairy industry worldwide (1, 2). The global economic loss due to mastitis was estimated to be $35 billion per year, including reduced milk production, condemnation of milk due to antibiotic residues, veterinary costs, culling of chronically infected cows, and occasional deaths (3). Moreover, mastitis poses a threat to human health since it may be responsible for zoonoses and for food toxin infections (4).

Mastitis is complex, developing as a result of the interaction between various factors associated with the host, specific pathogens, environment, and management (5). Over 200 different organisms have been recorded to cause bovine mastitis (6). Among the pathogenic bacteria, *Staphylococcus aureus* is one of the most common pathogens to cause bovine mastitis, especially in China (7–9). Antibiotic treatment is a key component for treatment of diseases caused by *S. aureus*. However, the abuse of antibiotics has led to the resistance of bacteria

Address correspondence to Hongsheng Li, lihsheng@sina.com.

The authors declare no conflict of interest.

**TABLE 1** Distribution of *S. aureus* isolates

| Province | No. of samples | No. of strains | Isolation rate |
|---|---|---|---|
| Gansu | 451 | 53 | 11.75% |
| Jiangsu | 237 | 11 | 4.64% |
| Shandong | 227 | 20 | 8.81% |
| Shanxi | 223 | 26 | 11.66% |
| Xinjiang | 216 | 25 | 11.57% |
| Heilongjiang | 195 | 18 | 9.23% |
| Ningxia | 182 | 15 | 8.24% |
| Inner Mongolia | 166 | 22 | 13.25% |
| Sichuan | 166 | 17 | 10.24% |
| Guizhou | 151 | 18 | 11.92% |
| Jilin | 147 | 11 | 7.48% |
| Hebei | 136 | 14 | 10.29% |
| Qinghai | 134 | 20 | 14.93% |
| Henan | 121 | 13 | 10.74% |
| Shaanxi | 112 | 6 | 5.36% |
| Hubei | 98 | 9 | 9.18% |
| In total | 2962 | 298 | 10.06% |

to drugs in recent years, and the issue of multidrug resistance has become increasingly prominent (10). There is a close relationship between bacterial resistance and drug resistance genes. *mecA* gene is considered one of the major resistance genes that confer resistance to $\beta$-lactams (11). In addition, *blaZ* gene also plays an important role in beta-lactam resistance in *S. aureus* (12).

At the same time, due to the toxicity and infectivity of *S. aureus*, the harm of *S. aureus* is stronger. This is related to the virulence genes the bacteria contain; *S. aureus* attaches to epithelial cells of the teat canal depending on the interaction of bacterial surface proteins, such as clumping factors A and B (*clfA* and *clfB*) and fibronectin-binding protein A (*fnbA*), providing assurance that the bacteria further infect the mammary gland (13, 14). Enterotoxin is an important virulence means of *S. aureus*; more than 20 different enterotoxins have been characterized, and these enterotoxins have been traditionally subdivided into classical genes (*sea* to *see*), which have been well distinguished and used in various detection methods. In addition, the toxic shock syndrome toxin TSST-1 and staphylococcal exfoliative toxin (ET) are important causes of mastitis with *S. aureus* (15, 16).

The aim of the present study was to investigate the relevant situation of *S. aureus* in dairy cow mastitis and to characterize these strains by the corresponding antibiotic drug sensitivity test, through evaluation of the genes of the drug resistance and the virulence by PCR molecular analyses.

## RESULTS

**Isolation and identification of *S. aureus* strains.** A total of 298 *S. aureus* strains were isolated from 2,962 mastitis samples as shown in Table 1.

**Antimicrobial susceptibility testing.** As shown in Table 2, the 298 *S. aureus* strains were tested for resistance to 11 antimicrobial agents by using the K-B disk diffusion method. Antimicrobial resistance was observed most frequently to penicillin G (91.95%), and all bacteria are resistant to at least two antibiotics (100%). However, all strains were sensitive to nitrofurantoin and sulfamethoxazole/trimethoprim.

**Bacterial typing.** The results of capsular polysaccharide typing test indicated that cp8 (57.38%) *S. aureus* was the most popular strain in China, followed by cp5 (35.57%) and cp336 (7.05%) as shown in Table 3.

A total of 48 spa types were identified within 298 strains. The most prevalent spa type was t459 (18.79%), followed by t6367 (16.78%), t067 (9.40%), t163 (6.38%), t4904 (4.03%), t13751 (3.69%), et al. (Table 4).

**Antimicrobial resistance genes.** In the experiments of antimicrobial resistance genes, the highest frequency gene we isolated was *blaZ* gene (92.95%), followed by *aacA-aphD* (87.25%), *tetK* (48.66%), *tetM* (27.85%), *norA* (55.03%), *norB* (53.36%), and *norC* (57.71%).

**TABLE 2** Results of 298 strains of *S. aureus* in drug resistance experiments

| Antibiotic class | Antibiotic | Strains, no. (%) |
|---|---|---|
| β-Lactam | Penicillin G | 274 (91.95) |
| | Cefoxitin | 77 (25.84) |
| Aminoglycoside | Gentamicin | 159 (53.86) |
| | Kanamycin | 194 (65.10) |
| Lincosamide | Clindamycin | 167 (56.04) |
| Quinolone | Ciprofloxacin | 146 (48.99) |
| | Levofloxacin | 146 (48.99) |
| Chloramphenicol | Chloramphenicol | 26 (8.72) |
| Nitrofuran | Nitrofurantoin | 0 (0) |
| Chain-positive bacteriocin | Quinupristin/dalfopristin | 3 (1.01) |
| Rifamycin | Rifampicin | 54 (18.12) |
| Tetracycline | Tetracycline | 73 (24.50) |
| Sulfonamide | Sulfamethoxazole/trimethoprim | 0 (0) |
| Oxazolidinone | Linezolid | 1 (0.34) |
| Multidrug resistant | | 298 (100) |

As a key gene for methicillin-resistant *S. aureus*, 73 strains of *mecA* gene were detected and confirmed by PBP2a gel experiment (Table 5).

**Detection of virulence determinants.** Among 298 strains of *S. aureus*, we detected genes such as adhesion factors, enterotoxin, toxic shock syndrome toxin, and exfoliative toxins, among which the *clfA* (97.99%) has the highest frequency, leaving the *clfB* (96.64%), *fnbA* (96.64%), *ebpS* (36.56%), *sea* (17.45%), *seb* (16.44%), *sec* (7.38%), *sed* (1.68%), *see* (0%), *tst* (23.50%), *eta* (1.34%), and *etb* (0%) (Table 6).

## DISCUSSION

*S. aureus* represents a major agent of contagious bovine mastitis. Our study indicated that 10.06% (298/2962) of mastitis samples were positive for *S. aureus*, which is significantly lower than previous reports by Liu et al. (17) and Zhang et al. (18) (27.7% and 29%). However, our data are similar to those of the report by Seo et al. (11.6%, 40/345) (19). In addition, 2% to 50% and even higher prevalence of *S. aureus* mammary gland infection was also observed in another report (20). Overall, *S. aureus* is still very common in China.

The resistance of *S. aureus* to antimicrobial agents is an increasing global problem. A drug sensitivity test is required not only for effective therapy but also for monitoring the spread of resistant strains. Eleven antimicrobial agents were used in this study, and all isolates were resistant to at least two antibiotics, especially to penicillin G (91.95%). Penicillin is a well-known antibiotic that is widely used in clinical practice, leading to the general resistance of *S. aureus*. We found that 91.95% of *S. aureus* isolates were resistant to penicillin, which was similar to earlier reports from China by Liu et al. (17), who reported that 85.2% of *S. aureus* isolates exhibited resistance to penicillin G, and Jian-Ning et al. (21), who reported 94.6%. However, Haran et al. (22) proved that 16% of 93 *S. aureus* isolates from 42 farms in America were resistant to penicillin. Another report from New Zealand in 2014 (23) found that of 364 *S. aureus* isolates, 28% were resistant to penicillin. Our data indicated that penicillin-resistant *S. aureus* in Chinese dairy farms is more common than that in other countries.

At the same time, we found that the resistance rates of kanamycin (65.10%), clindamycin (56.04%), gentamicin (53.86%), ciprofloxacin (48.99%), and levofloxacin (48.99%) were also very high. In contrast, these isolates had low drug resistance to quinupristin/dalfopristin (1.01%), linezolid (0.34%), nitrofurantoin (0%), and sulfamethoxazole/trimethoprim (0%). The clinical use of penicillin, kanamycin, clindamycin, ciprofloxacin, and levofloxacin should

**TABLE 3** Capsular polysaccharide typing of *S. aureus*

| Bacterial typing | Strains, no. (%) |
|---|---|
| cp5 | 106 (35.57) |
| cp8 | 171 (57.38) |
| cp336 | 21 (7.05) |

**TABLE 4** Spa typing of *S. aureus*

| Bacterial typing | Strains, no. (%) |
| --- | --- |
| t459 | 56 (18.79) |
| t6367 | 50 (16.78) |
| t458 | 3 (1.00) |
| t7073 | 9 (3.02) |
| t3932 | 10 (3.36) |
| t18401 | 3 (1.00) |
| t163 | 19 (6.38) |
| t304 | 3 (1.00) |
| t6811 | 5 (1.68) |
| t4904 | 12 (4.03) |
| t067 | 28 (9.40) |
| t9531 | 3 (1.00) |
| t6379 | 2 (0.67) |
| t2524 | 3 (1.00) |
| t3867 | 1 (0.34) |
| t4652 | 3 (1.00) |
| t7880 | 7 (2.35) |
| t3592 | 6 (2.01) |
| t17343 | 2 (0.67) |
| t521 | 6 (2.01) |
| t195 | 5 (1.68) |
| t6272 | 2 (0.67) |
| t3626 | 5 (1.68) |
| t527 | 7 (2.35) |
| t13751 | 11 (3.69) |
| t2246 | 1 (0.34) |
| t14061 | 1 (0.34) |
| t808 | 4 (1.34) |
| t1456 | 3 (1.00) |
| t1250 | 7 (2.35) |
| t4558 | 3 (1.00) |
| t9537 | 1 (0.34) |
| t14605 | 1 (0.34) |
| t1987 | 2 (0.67) |
| t12238 | 1 (0.34) |
| t870 | 1 (0.34) |
| t342 | 1 (0.34) |
| t10555 | 1 (0.34) |
| t1521 | 1 (0.34) |
| t14936 | 1 (0.34) |
| t5355 | 1 (0.34) |
| t6159 | 1 (0.34) |
| t079 | 1 (0.34) |
| t026 | 1 (0.34) |
| t233 | 1 (0.34) |
| t421 | 1 (0.34) |
| t3111 | 1 (0.34) |
| t4976 | 1 (0.34) |

be minimized. As nitrofurantoin was prohibited for use in food-producing animals in China, sulfamethoxazole/trimethoprim should be considered preferentially for the treatment of bovine mastitis caused by *S. aureus*. It is recommended to perform a drug sensitivity test before using an antibiotic drug (21).

The presence of resistance-associated genes in *S. aureus* was detected in this study. Resistance to penicillin is caused mainly by the *blaZ* gene encoding production of beta-lactamases, which hydrolytically destroy beta-lactams (24). An antimicrobial resistance genes test revealed that 92.95% of isolates were found carrying *blaZ* genes, agreeing with the finding that 91.95% of isolates were resistant to penicillin G. This is in agreement with data from Olsen et al. (25), showing that all penicillin-resistant strains carried *blaZ*. The activated *blaZ* could encode $\beta$-lactamase enzyme (penicillinase), which inactivates the antibiotic through

**TABLE 5** Status of antibiotic resistance-related genes in *S. aureus*

| Antibiotic | Gene | Strains, no. (%) |
|---|---|---|
| Tetracycline | *tetK* | 145 (48.66) |
| | *tetM* | 83 (27.85) |
| Aminoglycosides | *aacA-aphD* | 260 (87.25) |
| β-Lactams | *blaZ* | 277 (92.95) |
| | *mecA* | 73 (24.50) |
| | *norA* | 164 (55.03) |
| Quinolones | *norB* | 159 (53.36) |
| | *norC* | 172 (57.71) |

hydrolysis of the peptide bond in the β-lactam ring (26). Methicillin-resistant *Staphylococcus aureus* (MRSA) is a growing concern worldwide and has increasingly been recognized in farm animal populations in recent years (11, 22, 27). MRSA isolates are frequently multidrug resistant (MDR), which can result in higher costs, longer treatment times, and higher rates of hospitalization and comorbidities. Presence of *mecA* gene is generally recognized as the most reliable method for detection of methicillin resistance. In this study, 73 strains (24.50%) positive for the *mecA* gene were detected and confirmed by PBP2a gel experiment. However, some previous reports from China revealed lower isolation rates of MRSA with 15.53% (34/219) (28) and 14.20% (22/155) (29), which indicated the increasing presence of MRSA isolates in Chinese dairy cattle. Therefore, careful monitoring of the resistance status of *S. aureus* in dairy environments is needed, as the presence of MRSA poses potential risk to farm workers, veterinarians, and farm animals (17, 22).

Capsular polysaccharide serotyping of *S. aureus* was first reported in 1982 (30), and 11 serotypes have been described. Our results showed that cp8 (57.38%) *S. aureus* was the most popular strain in Chinese dairy farms, followed by cp5 (35.57%) and cp336 (7.05%). The existence of capsular polysaccharides in *S. aureus* plays an important role in the pathogenicity and immunogenicity. Our data provide a reference for the research and development of bovine mastitis vaccine. Molecular characterization of *S. aureus* is vital for the rapid identification of prevalent strains and will contribute to the control and prevention of *S. aureus*. Spa typing, which relies only on the assessment of the number of and sequence variation in repeats at the x region of the spa gene, exhibits excellent discriminatory power and has become a useful typing tool for the sake of its ease of performance, less expensive procedure, and standardized nomenclature (31). In our study, a total of 48 spa types were identified within 298 strains, of which t459 (18.79%) was the most prevalent spa type, followed by t6367 (16.78%), t067 (9.40%), t163 (6.38%), t4904 (4.03%), t13751 (3.69%), et al. A previous study reported that spa t224 (30.4%) was the most common type in Ningxia province of China (32). However, another study demonstrated that t267 (35.84%) was the predominant spa type in Liaoning province of China (33). Interestingly, t224 and t267 spa types were not detected at all in our study. Our results therefore suggest that the distribution of spa types varies between the different regions of China.

**TABLE 6** Detection of virulence determinants of *S. aureus*

| Toxin category | Gene | Strains, no. (%) |
|---|---|---|
| Adhesion factor | *fnbA* | 288 (96.64) |
| | *clfA* | 292 (97.99) |
| | *clfB* | 288 (96.64) |
| | *ebpS* | 109 (36.56) |
| Enterotoxin | *sea* | 52 (17.45) |
| | *seb* | 49 (16.44) |
| | *sec* | 22 (7.38) |
| | *sed* | 5 (1.68) |
| | *see* | 0 (0) |
| Toxic shock syndrome toxin | *tst* | 70 (23.50) |
| Exfoliative toxin | *eta* | 4 (1.34) |
| | *etb* | 0 (0) |

The broad range of infections caused by *S. aureus* is related to a number of virulence factors that allow it to adhere to surfaces, invade or avoid the immune system, and cause harmful toxic effects to the host (34). Adhesins are considered the most important virulence factors during early phases of *S. aureus* infection. The fibronectin-binding proteins (FnbP) and clumping factor proteins (*clfA*, *clfB*) can promote adhesion of *S. aureus* cells to a variety of molecules and surfaces, and they have been implicated in cell-cell adhesion. The *fnbA* and *clfB* genes were detected in almost all 298 isolates, which approaches the data reported by Wei et al. (35). Enterotoxin is a class of low-molecular-weight proteins with superantigen activity and is highly resistant to denaturation. In our research, *sea* gene (17.45%) was the most frequent, and the rest were *seb* (16.44%), *sec* (7.38%), and *sed* (1.68%). The *see* gene was not detected in any of the isolates. Some studies report the prevalence of the *sea* and *sed* genes in 10% and 7.50% of isolates (36), and another reports 53.30% prevalence for *sea*, 3.30% for *seb*, 50% for *sec*, 4% for *sed*, and 46.60% for *see* in isolates of *S. aureus* of mastitis milk (37). This indicates that the enterotoxin gene content of bacteria is different in different regions. Toxic shock syndrome toxin TSST-1 and staphylococcal toxin (ETs) are also an important part of *S. aureus* virulence. *tst* (23.50%) was also detected in a lower percentage than other genes in this study and was not found in the study reported in reference 38.

**Conclusions.** Our study indicated that 10.06% (298/2962) of mastitis samples from China were positive for *S. aureus*, and all isolates (100%) were resistant to at least two antibiotics, especially to penicillin G (91.95%). Sulfamethoxazole/trimethoprim should be considered preferentially for the treatment of bovine mastitis caused by *S. aureus*. Type 8 (57.38%) was the dominant capsular polysaccharide serotype, and t459 (18.79%) was the dominant spa type. The resistance genes of several antibiotics were detected, of which *blaZ* gene (92.95%) was highest, and the most prevalent virulence gene was *clfA* (98%).

## MATERIALS AND METHODS

**Collection of samples.** From 2016 to 2020, a total of 2,962 mastitis samples were collected from 43 large-scale dairy farms located in 16 provinces of China as shown in Table 1. Most of them ($n = 2,370$) were collected from clinical mastitis cows, and 592 mastitis samples were collected from subclinical mastitis cows. After disinfecting with 75% alcohol, collected milk samples were stored in cold storage and sent to the laboratory for bacterial isolation and identification.

**Isolation and identification.** Samples were inoculated in blood agar plates (Huankai Microbial Sci&Tech co., Ltd., Guangdong, China), and a typical single colony from each sample was picked for purification. Purified bacteria were subjected to smearing, Gram staining, and microscopic examination using the CHROM agar chromogenic medium for isolation and direct differentiation of *S. aureus* (Shanghai Central Bio-engineering Co., Ltd., Shanghai, China). Genetic testing was performed using a TakaLa 16S rRNA gene bacterial identification PCR kit (TaKaRa Biomedical Technology Co., Ltd., Dalian, China). Purified products were sent to the sequencing company, and the sequencing results were analyzed and determined by the BLAST program on the NCBI website.

**Antimicrobial resistance.** The drug sensitivity test was carried out according to the standards of the American Committee for Clinical Laboratory Standards using the K-B disk diffusion method (39). Penicillin G (10 $\mu$g), cefoxitin (30 $\mu$g), ciprofloxacin (5 $\mu$g), levofloxacin (5 $\mu$g), gentamicin (10 $\mu$g), kanamycin (30 $\mu$g), clindamycin (2 $\mu$g), chloramphenicol (30 $\mu$g), nitrofurantoin (30 $\mu$g), quinupristin/dalfopristin (15 $\mu$g), rifampin (5 $\mu$g), tetracycline (30 $\mu$g), sulfamethoxazole/trimethoprim (25 $\mu$g), and linezolid (30 $\mu$g) were used as antimicrobial agents (Oxoid, Basingstoke, UK).

**Bacterial typing.** Both capsular polysaccharide typing and spa typing were carried out on all strains. According to Verdier et al. (40), Cap5 k1 (5′-GTCAAAGATTATGTGATGCTACTGAG-3′), Cap5 k2 (5′-ACTTCGAATATA AACTTGAATCAATGTTATACAG-3′) located in cap5k for capsular type 5 and cap8 k1 (5′-GCCTTATGTTAGGTGA TAAACC-3′), cap8 k2 (5′-GGAAAAACACTATCATAGCAGG-3′) located in cap8I were synthesized. The polymorphic X region of the protein A gene (spa) was amplified using the primers spa-1113f (5′ TAA AGA CGA TCC TTC GGT GAG C 3′) and spa-1514r (5′ CAG CAG TAG TGC CGT TTG CTT 3′). The results were submitted to the website for further detection (https://www.spaserver.ridom.de) (41).

**Antimicrobial resistance genes.** According to the results of the drug sensitivity test, the related genes of several antibiotics with the highest drug resistance were selected. Penicillin (*mecA*, *blaZ*) (42), aminoglycosides (*aac*) (43), tetracycline (*tetK*, *tetM*) (14, 42), and quinolones (*norA*, *norB*, *norC*) (44) were detected by PCR. For *mecA*-positive strains, methicillin-resistant *S. aureus* (MRSA) was confirmed by PBP2a gel test.

**Detection of virulence determinants.** The genes encoding staphylococcal adhesion factor (*fnbA*, *clfA*, *clfB*, *ebpS*), enterotoxin (*sea*, *seb*, *sec*, *sed*, *see*), poisoning syndrome toxin (*tst*), and shedding toxin (*eta*, *etb*) were tested by PCR in this study (45, 46).

## ACKNOWLEDGMENTS

This research was funded by International Science and Technology Cooperation Project of Gansu Province (21YF5WA144), Research and Application of Green and Efficient Control Technology for Main Diseases of domestic animal (CAAS-LMY-02), the National Natural

Science Foundation of China (31802232), and the Agricultural Science and Technology Innovation Program (CAAS-ASTIP- 2014-LIHPS-03).

We declare no conflicts of interest.

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
