## [Reviewer comments · Microbiology Spectrum]

Microbiology Spectrum

Detection of Antibiotic Resistance, Virulence Gene and Drug Resistance Gene of *Staphylococcus aureus* Isolates from Bovine Mastitis

Zhe Zhang, Yun Chen, Xurong Wang, Xinpu Li, and Hongsheng Li

Corresponding Author(s): Hongsheng Li, Lanzhou Institute of Husbandry and Pharmaceutical Sciences of Chinese Academy of Agricultural Sciences

Review Timeline:

Submission Date:	February 14, 2022
Editorial Decision:	March 14, 2022
Revision Received:	May 18, 2022
Editorial Decision:	May 19, 2022
Revision Received:	June 7, 2022
Accepted:	June 9, 2022

Editor: Aude Ferran

Reviewer(s): Disclosure of reviewer identity is with reference to reviewer comments included in decision letter(s). The following individuals involved in review of your submission have agreed to reveal their identity: Yao-Hong Zhu (Reviewer #2); Chuang Xu (Reviewer #3)

Transaction Report:

DOI: <https://doi.org/10.1128/spectrum.00471-22>

March 14, 2022

Prof. Hongsheng Li
Lanzhou Institute of Husbandry and Pharmaceutical Sciences of Chinese Academy of Agricultural Sciences
Lanzhou
China

Re: Spectrum00471-22 (Detection of Antibiotic Resistance, Virulence Gene and Drug Resistance Gene of Staphylococcus aureus Isolates from Bovine Mastitis)

Dear Prof. Hongsheng Li,

Thank you for submitting your manuscript to Microbiology Spectrum. Your manuscript has been reviewed by 3 reviewers and some modifications of your manuscript are requested before publication. You will find the reviewer's comments at the end of this email and in attached files.

Link Not Available

Sincerely,

Aude Ferran

Journals Department
Reviewer comments:

Reviewer #1 (Comments for the Author):

The article 'Detection of Antibiotic Resistance, Virulence Gene and Drug Resistance Gene of Staphylococcus Aureus Isolates from Bovine Mastitis' is initially focusing on the AMR profile of S. aureus, which is the leading cause of mastitis in the bovine. The article is very well written except for some minor mistakes, which need correction (e.g. genetic testing was..... and sequencing was sent to.....) and several typo errors as well. As such, the article should be proofread carefully. While an

adequate number of samples have been analyzed. Overall, the article is based on a very basic analysis so there are some major concerns, which should be addressed;

1. The authors should clarify that a single colony per sample was analyzed?
2. What was the selection criteria for the selection of antibiotics for sensitivity testing? if it was based on the treatment of mastitis, then several of these antibiotics are not used in the treatment of mastitis.
3. In the sensitivity testing method is incomplete, authors should determine how they calculated the results, also add the pictures of the most susceptible antibiotics to show their zones.
4. The authors should justify the selection of virulence genes?
5. Lastly, and most importantly, the current data without statistical analysis is of no use. The authors apply suitable statistics and compare the data of resistance and virulence.
6. The conclusion of the study is not correctly supported by the results of the study.

Reviewer #3 (Comments for the Author):

The objective of this manuscript to was to detect antibiotic resistance, drug resistance gene and virulence gene of *S. aureus* strains. The research in this paper was comprehensive and the number of samples was large, which could reflect effectively the research results. This study will have important reference value for the prevention and control of cow mastitis. However, departmental language needs to be optimized, and the discussion of results is inadequate. My other views on the manuscript are as follows:

- (1) Line 40: delete " that cause mastitis in cows "
- (2) Lines 49-52: It should be rewritten.
- (3) Line 58: delete " field "
- (4) Line 69: In Figure 1, Heilong River should be Heilongjiang. I suggest to delete Figure 1 and put it in words.
- (5) Line 72, "Huankai Microbial Sci&Tech.co., Ltd."should be Huankai Microbial Sci&Tech Co., Ltd.
- (6) line 83-87 , 10ug, 30ug,5ug should be 10μg, 30μg, 5μg, etc.
- (7) Lines 109-116: I suggest modifying these sentences. For example, "...including 18 strains(9.23%) in 195 samples from Heilongjiang Province" may be better than"...including 18 strains(9.23%) in 195 Heilong River samples".
- (8) Discussion: Lacking discussion about Capsular polysaccharide and Spa types of *Staphylococcus aureus*.
- (9) In addition, the authors should keep two digits after the decimal point thought out this manuscript, and revise the References section.

Staff Comments:

Preparing Revision Guidelines

Please return the manuscript within 60 days; if you cannot complete the modification within this time period, please contact me. If you do not wish to modify the manuscript and prefer to submit it to another journal, please notify me of your decision immediately so that the manuscript may be formally withdrawn from consideration by Microbiology Spectrum.

Corresponding authors may join or renew ASM membership to obtain discounts on publication fees. Need to upgrade your

membership level? Please contact Customer Service at Service@asmusa.org.

The article 'Detection of Antibiotic Resistance, Virulence Gene and Drug Resistance Gene of Staphylococcus Aureus Isolates from Bovine Mastitis' is initially focusing on the AMR profile of S. aureus, which is the leading cause of mastitis in the bovine.

The article is very well written except for some minor mistakes, which need correction (e.g. genetic testing was..... and sequencing was sent to.....) and several typo errors as well. As such, the article should be proofread carefully. While an adequate number of samples have been analyzed. Overall, the article is based on a very basic analysis so there are some major concerns, which should be addressed;

1. The authors should clarify that a single colony per sample was analyzed?
2. What was the selection criteria for the selection of antibiotics for sensitivity testing? if it was based on the treatment of mastitis, then several of these antibiotics are not used in the treatment of mastitis.
3. In the sensitivity testing method is incomplete, authors should determine how they calculated the results, also add the pictures of the most susceptible antibiotics to show their zones.
4. The authors should justify the selection of virulence genes?
5. Lastly, and most importantly, the current data without statistical analysis is of no use. The authors apply suitable statistics and compare the data of resistance and virulence.
6. The conclusion of the study is not correctly supported by the results of the study.

Dear Reviewers:

Thank you for your comments concerning our manuscript entitled “Detection of Antibiotic Resistance, Virulence Gene and Drug Resistance Gene of *Staphylococcus Aureus* Isolates from Bovine Mastitis” (Spectrum00471-22), which are very valuable and helpful for revising and improving our paper. We have studied comments carefully and have done appropriately modified on the manuscript. Those changes are highlighted in revised manuscript and the responds to the comments are as flowing:

Response to Reviewer #1:

The article is very well written except for some minor mistakes, which need correction (e.g. genetic testing was..... and sequencing was sent to.....) and several typo errors as well. As such, the article should be proofread carefully. While an adequate number of samples have been analyzed. Overall, the article is based on a very basic analysis so there are some major concerns, which should be addressed;

Response: We have carefully checked and improved the English writing in the revised manuscript, and asked a professor, who is a well established expert, to polish our paper.

1. The authors should clarify that a single colony per sample was analyzed?

Response: We have clarified this issue in line 75 of revised version.

2. What was the selection criteria for the selection of antibiotics for sensitivity testing? if it was based on the treatment of mastitis, then several of these antibiotics are not used in the treatment of mastitis.

Response: Generally speaking, it is appropriate to choose the antibiotics commonly used in the treatment of mastitis. However, the *Staphylococcus aureus* is zoonotic pathogen, a wide range of antibiotics were chose to find out whether these pathogens are resistant to human drugs.

3. In the sensitivity testing method is incomplete, authors should determine how they calculated the results, also add the pictures of the most susceptible antibiotics to show their zones.

Response: Pictures and criteria for drug sensitivity test have been added in the revised version.

4. The authors should justify the selection of virulence genes?

Response: As we know, *Staphylococcus aureus* have plenty of virulence factors, including a vast ability to evade host immune defenses. Several important and common virulence factors were tested by PCR in this study.

5. Lastly, and most importantly, the current data without statistical analysis is of no use. The authors apply suitable statistics and compare the data of resistance and virulence.

Response: Thanks for your pertinent suggestion. The main purpose of this study is to investigate the prevalent status of resistance and virulence of *Staphylococcus aureus* in dairy farm, and the correlation between resistance and virulence was not considered at the beginning of experiment design. Thank you again for your pertinent suggestion, and we will focus on this problem in the future study.

6. The conclusion of the study is not correctly supported by the results of the study.

Response: We have modified the conclusion in the revised version.

Response to Reviewer #2:

Majors:

1. The sample size of this study is huge. However, this study did not link drug sensitivity with capsular polysaccharide and antibiotic resistance related genes. We do not know if an isolate which is drug sensitivity to an antibiotic is equipped with this antibiotic resistance-related gene. We do not know if capsular polysaccharide and antibiotic resistance-related genes have an effect on drug resistance.

Response:

Thank you for your professional advice.

There has been a long debate about the relationship between resistance genes and resistance phenotypes, since a strain is resistant to an antibiotic and often carries the corresponding resistance genes. Conversely, if a strain carries a specific resistance gene, it does not necessarily show resistance to antibiotics.

The main purpose of this study is to investigate the prevalent status of resistance and virulence of *Staphylococcus aureus* in dairy farm, and the correlation between drug sensitivity, virulence factors and related genes was not considered at the beginning of experiment design. Discussions about these these problems according to the existing data were added in revised manuscript, and we will pay more attention to these problems in the future study.

2. The author only shows the isolation rate of virulence genes, but does not introduce the importance of these detected virulence genes to *Staphylococcus aureus* and the harm to the host.

Response: Relevant discussions about virulence factors have been added in the revised version.

3. This manuscript should be extensive editing of English language.

Response: We have carefully checked and improved the English writing in the revised manuscript, and asked a professor, who is a well established expert, to polish our paper.

Minors:

1. Line 24, “...capsular polysaccharide serotype, t459 was the dominant spa type.” Should be “...capsular polysaccharide serotype and t459 was the dominant spa type.”

Response: We have corrected this sentence.

2. The name of genes should be italic.

Response: All genes have corrected in revised version.

3. Line 27, “...antimicrobial resistance, virulence genes of...” should be “...antimicrobial resistance and virulence genes of...”

Response: We have corrected this sentence.

4. Line 49, omit “made.”

Response: We have delete “made” from revised version.

5. In Figure 1, “Heilong River” should be “Heilongjiang.” “Heinan” should be “Henan.”

Response: We have corrected the errors in Figure 1.

6. There should be a space between a number and its unit, such as “10ug” in Section 2.3.

Response: We have added space between a number and its unit in full text.

7. Line 100, “penicillin” should be “Penicillin.”

Response: We have corrected this error in revised version.

8. Line 108, omit “the text continues here.” ?

Response: We have delete “the text continues here” from revised version.

9. Line 110, “Heilong River” should be “Heilongjiang.” Line 113, “Heinan” should be “Henan.”

Response: We have corrected this error in revised version.

9. There is a space before brackets.

Response: We have added space before brackets in revised version.

10. Rewrite Section 3.1, such as 18 strains out of 195 samples in Heilongjiang? The periods should be replaced with commas, such as “Guizhou samples. 6 strains...”

Response: We have rewritten Section 3.1 in revised version.

12. Use the same font, such as Line 154.

Response: The format of revised version has been modified.

Response to Reviewer #3:

The objective of this manuscript was to detect antibiotic resistance, drug resistance gene and virulence gene of *S. aureus* strains. The research in this paper was comprehensive and the number of samples was large, which could reflect effectively the research results. This study will have important reference value for the prevention and control of cow mastitis. However, departmental language needs to be optimized, and the discussion of results is inadequate.

Response: Thank you for your affirmation of our paper.

We have carefully checked and improved the English writing in the revised manuscript, and asked a professor, who is a well established expert, to polish our paper.

The discussion and conclusion part of the manuscript have been rewritten.

My other views on the manuscript are as follows:

(1) Line 40: delete " that cause mastitis in cows "

Response: " that cause mastitis in cows " has been deleted in the revised version.

(2) Lines 49-52: It should be rewritten.

Response: We have rewritten related paragraph.

(3) Line 58: delete " field "

Response: " field " has been deleted.

(4) Line 69: In Figure 1, Heilong River should be Heilongjiang. I suggest to delete Figure 1 and put it in words.

Response: We have corrected the errors in Figure 1.

(5) Line 72, "Huankai Microbial Sci&Tech.co., Ltd." should be Huankai Microbial Sci&Tech Co., Ltd.

Response: We have corrected the mistake in revised version.

(6) line 83-87, 10ug, 30ug, 5ug should be 10 μ g, 30 μ g, 5 μ g, etc.

Response: We have corrected these mistakes in revised version.

(7) Lines 109-116: I suggest modifying these sentences. For example, "...including 18 strains(9.23%) in 195 samples from Heilongjiang Province" may be better than "...including 18 strains(9.23%) in 195 Heilong River samples".

Response: We have rewritten this part in revised version.

(8) Discussion: Lacking discussion about Capsular polysaccharide and Spa types of *Staphylococcus aureus*.

Response: The discussion about Capsular polysaccharide and Spa types have been added in revised version.

(9) In addition, the authors should keep two digits after the decimal point throughout this manuscript, and revise the References section.

Response: We have corrected these mistakes in revised version.

May 19, 2022

Prof. Hongsheng Li
Lanzhou Institute of Husbandry and Pharmaceutical Sciences of Chinese Academy of Agricultural Sciences
Lanzhou
China

Re: Spectrum00471-22R1 (Detection of Antibiotic Resistance, Virulence Gene and Drug Resistance Gene of *Staphylococcus aureus* Isolates from Bovine Mastitis)

Dear Prof. Hongsheng Li:

Thank you for submitting your revised manuscript to Microbiology Spectrum.

Some few modifications are required (I attached the manuscript with points to revise highlighted)

- Table2 is entitled "sensitivity" while the described results concern resistance. It is not easy understandable whether the percentages refer to susceptibility or resistance.

- there is a mistake in the word "sulfamethoxazole" at different places in the text.

-you recommend nitrofurantoin but for me, it is carcinogenic and not allowed in food-producing animals. Please provide some arguments to recommend it or modify your text.

When submitting the revised version of your paper, please provide (1) point-by-point responses " not in your cover letter, and (2) a PDF file that indicates the changes from the original submission (by highlighting or underlining the changes) as file type "Marked Up Manuscript - For Review Only". Please use this link to submit your revised manuscript - we strongly recommend that you submit your paper within the next 60 days or reach out to me. Detailed instructions on submitting your revised paper are below.

Link Not Available

Sincerely,

Aude Ferran

Journals Department
Reviewer comments:

Staff Comments:

Preparing Revision Guidelines

Please return the manuscript within 60 days; if you cannot complete the modification within this time period, please contact me. If you do not wish to modify the manuscript and prefer to submit it to another journal, please notify me of your decision immediately so that the manuscript may be formally withdrawn from consideration by Microbiology Spectrum.

Dear Reviewers:

Thank you for your comments concerning our manuscript entitled “Detection of Antibiotic Resistance, Virulence Gene and Drug Resistance Gene of *Staphylococcus Aureus* Isolates from Bovine Mastitis” (Spectrum00471-22), which are very valuable and helpful for revising and improving our paper. We have studied comments carefully and have done appropriately modified on the manuscript. Those changes are highlighted in revised manuscript and the responds to the comments are as flowing:

Response to Reviewer #1:

The article is very well written except for some minor mistakes, which need correction (e.g. genetic testing was..... and sequencing was sent to.....) and several typo errors as well. As such, the article should be proofread carefully. While an adequate number of samples have been analyzed. Overall, the article is based on a very basic analysis so there are some major concerns, which should be addressed;

Response: We have carefully checked and improved the English writing in the revised manuscript, and asked a professor, who is a well established expert, to polish our paper.

1. The authors should clarify that a single colony per sample was analyzed?

Response: We have clarified this issue in line 75 of revised version.

2. What was the selection criteria for the selection of antibiotics for sensitivity testing? if it was based on the treatment of mastitis, then several of these antibiotics are not used in the treatment of mastitis.

Response: Generally speaking, it is appropriate to choose the antibiotics commonly used in the treatment of mastitis. However, the *Staphylococcus aureus* is zoonotic pathogen, a wide range of antibiotics were chose to find out whether these pathogens are resistant to human drugs.

3. In the sensitivity testing method is incomplete, authors should determine how they calculated the results, also add the pictures of the most susceptible antibiotics to show their zones.

Response: Pictures and criteria for drug sensitivity test have been added in the revised version.

4. The authors should justify the selection of virulence genes?

Response: As we know, *Staphylococcus aureus* have plenty of virulence factors, including a vast ability to evade host immune defenses. Several important and common virulence factors were tested by PCR in this study.

5. Lastly, and most importantly, the current data without statistical analysis is of no use. The authors apply suitable statistics and compare the data of resistance and virulence.

Response: Thanks for your pertinent suggestion. The main purpose of this study is to investigate the prevalent status of resistance and virulence of *Staphylococcus aureus* in dairy farm, and the correlation between resistance and virulence was not considered at the beginning of experiment design. Thank you again for your pertinent suggestion, and we will focus on this problem in the future study.

6. The conclusion of the study is not correctly supported by the results of the study.

Response: We have modified the conclusion in the revised version.

Response to Reviewer #2:

Majors:

1. The sample size of this study is huge. However, this study did not link drug sensitivity with capsular polysaccharide and antibiotic resistance related genes. We do not know if an isolate which is drug sensitivity to an antibiotic is equipped with this antibiotic resistance-related gene. We do not know if capsular polysaccharide and antibiotic resistance-related genes have an effect on drug resistance.

Response:

Thank you for your professional advice.

There has been a long debate about the relationship between resistance genes and resistance phenotypes, since a strain is resistant to an antibiotic and often carries the corresponding resistance genes. Conversely, if a strain carries a specific resistance gene, it does not necessarily show resistance to antibiotics.

The main purpose of this study is to investigate the prevalent status of resistance and virulence of *Staphylococcus aureus* in dairy farm, and the correlation between drug sensitivity, virulence factors and related genes was not considered at the beginning of experiment design. Discussions about these these problems according to the existing data were added in revised manuscript, and we will pay more attention to these problems in the future study.

2. The author only shows the isolation rate of virulence genes, but does not introduce the importance of these detected virulence genes to *Staphylococcus aureus* and the harm to the host.

Response: Relevant discussions about virulence factors have been added in the revised version.

3. This manuscript should be extensive editing of English language.

Response: We have carefully checked and improved the English writing in the revised manuscript, and asked a professor, who is a well established expert, to polish our paper.

Minors:

1. Line 24, “...capsular polysaccharide serotype, t459 was the dominant spa type.” Should be “...capsular polysaccharide serotype and t459 was the dominant spa type.”

Response: We have corrected this sentence.

2. The name of genes should be italic.

Response: All genes have corrected in revised version.

3. Line 27, “...antimicrobial resistance, virulence genes of...” should be “...antimicrobial resistance and virulence genes of...”

Response: We have corrected this sentence.

4. Line 49, omit “made.”

Response: We have delete “made” from revised version.

5. In Figure 1, “Heilong River” should be “Heilongjiang.” “Heinan” should be “Henan.”

Response: We have corrected the errors in Figure 1.

6. There should be a space between a number and its unit, such as “10ug” in Section 2.3.

Response: We have added space between a number and its unit in full text.

7. Line 100, “penicillin” should be “Penicillin.”

Response: We have corrected this error in revised version.

8. Line 108, omit “the text continues here.” ?

Response: We have delete “the text continues here” from revised version.

9. Line 110, “Heilong River” should be “Heilongjiang.” Line 113, “Heinan” should be “Henan.”

Response: We have corrected this error in revised version.

9. There is a space before brackets.

Response: We have added space before brackets in revised version.

10. Rewrite Section 3.1, such as 18 strains out of 195 samples in Heilongjiang? The periods should be replaced with commas, such as “Guizhou samples. 6 strains...”

Response: We have rewritten Section 3.1 in revised version.

12. Use the same font, such as Line 154.

Response: The format of revised version has been modified.

Response to Reviewer #3:

The objective of this manuscript was to detect antibiotic resistance, drug resistance gene and virulence gene of *S. aureus* strains. The research in this paper was comprehensive and the number of samples was large, which could reflect effectively the research results. This study will have important reference value for the prevention and control of cow mastitis. However, departmental language needs to be optimized, and the discussion of results is inadequate.

Response: Thank you for your affirmation of our paper.

We have carefully checked and improved the English writing in the revised manuscript, and asked a professor, who is a well established expert, to polish our paper.

The discussion and conclusion part of the manuscript have been rewritten.

My other views on the manuscript are as follows:

(1) Line 40: delete " that cause mastitis in cows "

Response: " that cause mastitis in cows " has been deleted in the revised version.

(2) Lines 49-52: It should be rewritten.

Response: We have rewritten related paragraph.

(3) Line 58: delete " field "

Response: " field " has been deleted.

(4) Line 69: In Figure 1, Heilong River should be Heilongjiang. I suggest to delete Figure 1 and put it in words.

Response: We have corrected the errors in Figure 1.

(5) Line 72, "Huankai Microbial Sci&Tech.co., Ltd." should be Huankai Microbial Sci&Tech Co., Ltd.

Response: We have corrected the mistake in revised version.

(6) line 83-87, 10ug, 30ug, 5ug should be 10 μ g, 30 μ g, 5 μ g, etc.

Response: We have corrected these mistakes in revised version.

(7) Lines 109-116: I suggest modifying these sentences. For example, "...including 18 strains(9.23%) in 195 samples from Heilongjiang Province" may be better than "...including 18 strains(9.23%) in 195 Heilong River samples".

Response: We have rewritten this part in revised version.

(8) Discussion: Lacking discussion about Capsular polysaccharide and Spa types of *Staphylococcus aureus*.

Response: The discussion about Capsular polysaccharide and Spa types have been added in revised version.

(9) In addition, the authors should keep two digits after the decimal point throughout this manuscript, and revise the References section.

Response: We have corrected these mistakes in revised version.

Dear Editor:

Thank you for your comments concerning our manuscript (Spectrum00471-22). We have studied comments carefully and have done appropriately modified on the manuscript.

1、 Table2 is entitled "sensitivity" while the described results concern resistance. It is not easy understandable whether the percentages refer to susceptibility or resistance.

Response: We have corrected this mistake.

2、 there is a mistake in the word "sulfamethoxazole" at different places in the text.

Response: We have corrected this mistake in revised version.

3、 you recommend nitrofurantoin but for me, it is carcinogenic and not allowed in food-producing animals. Please provide some arguments to recommend it or modify your text.

Response: Thanks so much for your pertinent suggestion. Related sentences have modified in revised version.

June 9, 2022

Prof. Hongsheng Li
Lanzhou Institute of Husbandry and Pharmaceutical Sciences of Chinese Academy of Agricultural Sciences
Lanzhou
China

Re: Spectrum00471-22R2 (Detection of Antibiotic Resistance, Virulence Gene and Drug Resistance Gene of *Staphylococcus aureus* Isolates from Bovine Mastitis)

Dear Prof. Hongsheng Li:

Your manuscript has been accepted, and I am forwarding it to the ASM Journals Department for publication. You will be notified when your proofs are ready to be viewed.

Sincerely,

Aude Ferran
Editor, Microbiology Spectrum
